# Overexpression of Tfap2a in Mouse Oocytes Impaired Spindle and Chromosome Organization

**DOI:** 10.3390/ijms232214376

**Published:** 2022-11-19

**Authors:** Juan Lin, Zhuqing Ji, Zhengyang Di, Yeqing Zhang, Chen Yan, Shenming Zeng

**Affiliations:** National Engineering Laboratory for Animal Breeding, Key Laboratory of Animal Genetics, Breeding, and Reproduction of the Ministry of Agriculture and Rural Affairs, College of Animal Science and Technology, China Agricultural University, Beijing 100193, China

**Keywords:** Tfap2a, histone acetylation, histone lactylation, meiotic maturation, mouse oocyte

## Abstract

Transcription factor AP-2-alpha (Tfap2a) is an important sequence-specific DNA-binding protein that can regulate the transcription of multiple genes by collaborating with inducible viral and cellular enhancer elements. In this experiment, the expression, localization, and functions of Tfap2a were investigated in mouse oocytes during maturation. Overexpression via microinjection of Myc-Tfap2a mRNA into the ooplasm, immunofluorescence, and immunoblotting were used to study the role of Tfap2a in mouse oocyte meiosis. According to our results, Tfap2a plays a vital role in mouse oocyte maturation. Levels of Tfap2a in GV oocytes of mice suffering from type 2 diabetes increased considerably. Tfap2a was distributed in both the ooplasm and nucleoplasm, and its level gradually increased as meiosis resumption progressed. The overexpression of Tfap2a loosened the chromatin, accelerated germinal vesicle breakdown (GVBD), and blocked the first polar body extrusion 14 h after maturation in vitro. The width of the metaphase plate at metaphase I stage increased, and the spindle and chromosome organization at metaphase II stage were disrupted in the oocytes by overexpressed Tfap2a. Furthermore, Tfap2a overexpression dramatically boosted the expression of p300 in mouse GV oocytes. Additionally, the levels of pan histone lysine acetylation (Pan Kac), histone H4 lysine 12 acetylation (H4K12ac), and H4 lysine 16 acetylation (H4K16ac), as well as pan histone lysine lactylation (Pan Kla), histone H3 lysine18 lactylation (H3K18la), and H4 lysine12 lactylation (H4K12la), were all increased in GV oocytes after Tfap2a overexpression. Collectively, Tfap2a overexpression upregulated p300, increased the levels of histone acetylation and lactylation, impeded spindle assembly and chromosome alignment, and ultimately hindered mouse oocyte meiosis.

## 1. Introduction

In mammals, oocytes undergo the last round of DNA replication and initiate meiosis during fetal development; then, the oocytes are arrested at the diplotene stage of the first meiotic prophase inside ovarian follicles [1]. During this stage, the nucleus of the oocyte is traditionally termed the germinal vesicle (GV), which exhibits a special chromatin configuration that changes from a decondensed configuration (termed the non-surrounded nucleolus, NSN) into a condensed configuration around the nucleolus (termed the surrounded nucleolus, SN). There is also an intermediate type called IN. Usually, NSN-type oocytes exhibit high transcriptional activity [2] and are incapable of developing beyond the two-cell stage, whereas SN-type oocytes are transcriptionally silent [3] and can develop into the blastocyst stage [4,5]. The GV-block is maintained until a surge of luteinizing hormone initiates meiotic resumption [6]. After meiosis recovery, oocytes undergo a series of sequential changes, including chromatin condensation, nucleolar disintegration, GV rupture, and assembly of the spindle at the first meiotic metaphase [7]. Extrusion of the first polar body (PB1) marks the completion of the first meiosis. Subsequently, the oocyte enters second meiosis and is again arrested at the metaphase stage of the second meiosis (metaphase II, MII) until fertilization [8,9].

Mammalian oocytes are particularly error-inclined through the two successive meiotic divisions [10,11]. Factors such as diabetes [12,13] and advancing maternal age [14,15] can affect meiosis and are thought to mainly result in aneuploidy. As a major global health threat, type 2 diabetes mellitus (T2D) accounts for more than 90% of diabetes mellitus cases globally [16,17], which is closely linked to the epidemic of obesity and polycystic ovary syndrome [18] and can lead to ovulatory dysfunction and infertility. Among numerous genes known to be associated with T2D risk, mRNA expression of *TFAP2A* was higher in skeletal muscle tissues [19], blood [20], and liver [21] of patients suffering from T2D or a prediabetic phenotype identified as insulin-resistant.

As a class of proteins that bind to unique DNA sequences, the family of transcription factor activating protein-2 (Tfap-2) contains five representative members: α, β, γ, δ, and ε [22]. Members of the Tfap-2 family are characterized by a basic helix–span–helix domain and can bind to G/C-rich DNA consensus sequences, such as CCN3/4GGC, GCCN3/4GGG [23], or CCCCAGGC [24], and stimulate target gene transcription [25,26]. Tfap2a (also known as Tfap2α, which is encoded by the *Tfap2a* gene) can interact (via the N terminus) with p300 (via the N terminus) to mediate transcriptional activation [27]. As a histone acetyltransferase (HAT), p300 can transfer acetyl to the ε-NH3^+^ of a specific lysine at the amino terminus of core histones, neutralizing the positive charge of the histone tails and lowering their affinity to DNA, thus facilitating the accessibility of chromatin for transcription [28,29]. As a novel epigenetic modification, histone lysine lactylation was recently reported to directly stimulate gene transcription from chromatin [30] and some non-histone proteins [31,32] and was found to play important roles in numerous cellular processes [30,33,34]. Acetyltransferase p300 is the only reported potential writer of histone lactylation to date [30].

Multiple post-translational modifications occur during oocyte growth and maturation, including but not limited to the acetylation, methylation, phosphorylation, ubiquitination, and SUMOylation of various proteins [35,36,37,38]. These modifications are more or less involved in the regulation of the chromatin structure and gene expression. It is now well-known that histone acetylation is reversible and related to increased transcriptional activity in eukaryotic cells [39,40]. The reactions associated with the acetylation and deacetylation of lysine residues of histone are catalyzed by enzymes within the HAT family and the histone deacetylase (HDAC) family [41,42]. During oocyte growth and maturation, the level of histone acetylation changes dynamically [43,44]. Changes in histone lactylation during oocyte growth and maturation have been rarely studied in depth. 

*Tfap2a* can be induced by factors such as retinoic acid [45], UVA radiation, and singlet oxygen [46], and there is a high level of the *TFAP2A* gene in patients suffering from gestational diabetes mellitus and type 1 diabetes [47,48]. Overexpression of TFAP2A has been found in malignant epithelial ovarian tumors [49], nasopharyngeal carcinoma cells [50], and lung carcinoma [51] and is highly associated with poor prognosis. Apart from the level, the expression priority of Tfap2a in the nucleus rather than in the cytoplasm is related to increased risk of dying [49,52]. Maternal diabetes hinders the maturation of oocytes [53,54,55], but whether the level of Tfap2a changes in oocytes from diabetic mice remains to be clarified, as do the effects of the overexpression of Tfap2a on mouse oocyte maturation and histone posttranslational modifications in GV oocytes. In the present study, by employing mice with induced T2D, overexpression analysis, quantitative real-time polymerase chain reaction (qRT-PCR), and immunofluorescence staining, we investigated the expression and localization of Tfap2a during mouse oocyte meiosis and found that Tfap2a overexpression (Tfap2a-OE) regulates oocyte maturation, histone acetylation and lactylation, and the expression of p300.

## 2. Results

### 2.1. T2D Caused Spindle and Chromosome Defects in MII Oocytes and High Levels of Tfap2a in GV Oocytes

To confirm the effect of T2D on the meiosis of oocytes, we investigated the spindle and chromosome morphologies in the MII oocytes of normal and diabetic mice. As shown in Figure 1A, normal MII oocytes presented typical barrel-shaped bipolar spindles, with chromosomes well-aligned on the metaphase plate (Figure 1A(a)), whereas MII oocytes from diabetic mice revealed spindle defects and chromosome misalignment (Figure 1A(b–e)). Moreover, quantitative analysis demonstrated that the rates of spindle defects and misaligned chromosomes in the diabetic MII oocytes (74.8 ± 11.2% and 35.8 ± 5.6%) differed significantly from those in the control oocytes (32.5 ± 1.5% and 21.4 ± 5.7%) (*p* < 0.01 and *p* < 0.05) (Figure 1B). As shown in Figure 1C, the expression of *Tfap2a* mRNA sharply increased after diabetic induction (10,736 ± 4608% vs. 100 ± 30%, *p* < 0.01). The immunofluorescence staining results showed that the Tfap2a protein was localized in the nucleoplasm and cytoplasm of GV oocytes in both the diabetic and control mice, and the level of this protein in diabetic oocytes was significantly increased relative to that in the control group (Figure 1D). Overall, these results indicate that the level of Tfap2a in diabetic mouse oocytes was higher than that in the control group, which could be associated with the poor quality of oocytes from diabetic mice.

### 2.2. Expression and Subcellular Localization of Tfap2a during Mouse Oocyte Meiotic Maturation

As shown in Figure 2A, the mRNA level of *Tfap2a* gradually increased from the GV stage to the MII stage and reached the highest level at the MII stage. The mRNA levels of *Tfap2a* at the GVBD, metaphase I (MI), and MII stages were 203 ± 27, 199 ± 46, and 266 ± 57% of those at the GV stage, respectively. The results presented in Figure 2B show that in GV oocytes, Tfap2a was concentrated primarily in the nucleoplasm and ooplasm. After GVBD, Tfap2a was distributed in the cytoplasm from the GVBD stage to the MII stage. Consistent with the qRT-PCR results, the level of Tfap2a protein gradually increased from the GV stage to the MII stage in oocytes (Figure 2C). 

### 2.3. Tfap2a-OE Caused Chromatin Loosening, Premature GVBD, Blocked PB1 Extrusion, Chromosome Misalignment, and Spindle Disruption in Mouse Oocytes

As shown in Figure 3A, after mRNA injection and the incubation of oocytes for 12 h with 200 nM 3-isobutyl-1-methylxanthine (IBMX), the *Tfap2a* mRNA level was significantly higher in the Myc-Tfap2a group than that in the Myc group (6,787,710 ± 441,927% vs. 100 ± 11%, *p* < 0.0001). Western blotting was conducted to confirm Myc-Tfap2a fusion protein expression after exogenous mRNA injection (Figure 3B). Only one band (exogenous Myc-Tfap2a) was detected when the anti-Myc antibody was used in the injection group, whereas the Myc-Tfap2a group showed two bands (the above band was Myc-Tfap2a, and the bottom band was Tfap2a) when the blot was incubated with the anti-Tfap2a antibody. Immunofluorescence staining detected with anti-Myc antibody showed that Myc and Myc-Tfap2a mRNA were successfully translated to the protein, and Myc-Tfap2a was mainly colocalized with the DNA, with a small amount in the nucleoplasm. When anti-Tfap2a antibody was used, total Tfap2a in the nucleus was boosted and colocalized with the DNA in the Tfap2a-OE group, in contrast to the dispersed Tfap2a in the Myc group. Apart from NSN-oocytes, which presented dispersed chromatin, chromatin in the IN- and SN-type oocytes of the Tfap2a-OE group seemed more loosened, and the chromatin ring was thinner than that in the control oocytes (Figure 3C). 

As shown in Figure 4, after 3 h culture of in vitro maturation (IVM), the GVBD rate in the Myc-Tfap2a group was significantly increased relative to that in the Myc group (82.1 ± 5.2% vs. 64.0 ± 4.9%, *p* < 0.001). Although the rate of GVBD in the Myc-Tfap2a group presented higher levels than that in the Myc group throughout the entire process of IVM, the PB1 extrusion rate after 14 h culture in the Myc-Tfap2a group was remarkably lower than that in the Myc group (38.8 ± 8.6% vs. 75.7 ± 4.8%, *p* < 0.0001).

We then assessed the spindle and chromosome morphologies in the MII oocytes. The results showed that most oocytes in the Myc group exhibited a typical barrel-shaped spindle and well-aligned chromosomes at the equatorial plate (Figure 5A(a)). However, in the Tfap2a-OE group, severe chromosome misalignment was observed, with numerous chromosomes out of the equatorial plate (Figure 5A(b–d)). Some chromosomes were scattered in the cytoplasm far away from the nuclear region (Figure 5A(b)). Furthermore, unattached microtubules and elongated (Figure 5A(c)) or multipolar (Figure 5A(d)) spindles were observed. Quantitative analysis demonstrated that the rates of spindle defects and misaligned chromosomes in the Tfap2a-OE oocytes (96.5 ± 4.4% and 100.0 ± 0.0%) and control oocytes (32.0 ± 11.9% and 20.3 ± 6.2%) differed significantly (*p* < 0.001 and *p* < 0.0001) (Figure 5B). After culturing the Myc-overexpression oocytes for 8 h, chromosomes were concentrated at the mid-plate, whereas in the Tfap2a-OE group, the metaphase plate width (axis-to-axis distance between the two lines at the DNA edges) was significantly longer (27.4 ± 10.2 μm vs. 6.7 ± 3.0 μm, *p* < 0.0001, Figure 5C,D). Furthermore, it was obvious that the Myc-Tfap2a fusion protein was codistributed with condensed chromosomes in the MII oocytes after Tfap2a-OE (Figure 5A), in contrast to the endogenous Tfap2a shown in the control group (Figure 2B).

Taken together, these results show that the overexpression of Tfap2a may decondense the chromatin in GV oocytes, disrupt chromosome alignment and spindle assembly during mouse oocyte meiosis, and eventually hinder the maturation of oocytes.

### 2.4. Tfap2a-OE Caused Increased Levels of p300 in Mouse Oocytes at the GV Stage

As shown in Figure 6A, the immunofluorescence signals of p300 were concentrated in the nucleus. The quantitative fluorescence intensity data also showed that the level of p300 was markedly elevated in Tfap2a-OE oocytes relative to the control cells (Figure 6B). 

### 2.5. Tfap2a-OE Caused Increased Levels of Histone Acetylation and Lactylation in Mouse Oocytes at the GV Stage

We then assessed the effects of enhanced p300 after Tfap2a-OE on the histone posttranslational modifications (acetylation and lactylation) in GV oocytes. 

As shown in Figure 7A–C and Figure 8A–C, the levels of Pan Kac, H4K12ac, and H4K16ac in oocytes with NSN- and IN-chromatin configurations and acetylation at certain sites (H4K12 and H4K16) in SN-configuration oocytes were all increased significantly after Tfap2a-OE.

Both immunofluorescence images (Figure 7A,D,E) and the corresponding quantitative relative fluorescence intensity data (Figure 8D–F) showed that the levels of Pan Kla, H3K18la, and H4K12la were obviously higher in Tfap2a-OE oocytes than those in the Myc group.

The above data collectively confirm the positive role of Tfap2a-OE in histone acetylation and lactylation modifications in mouse GV oocytes. 

## 3. Discussion

As a transcription factor, Tfap2a participates in multiple biological processes, such as apoptosis [56], cell population proliferation [57,58], and the development of multiple organs [59,60,61]. In this study, we investigated, for the first time, the expression characteristics and potential functions of Tfap2a during mouse oocyte meiotic maturation. Abnormally high levels of Tfap2a caused by maternal diabetes might have contributed to the diminished oocyte quality. Overexpression of Tfap2a yielded loose chromatin and severe spindle and chromosome defects, accompanied by boosted p300 protein, histone acetylation and lactylation, and, eventually, accelerated GVBD and impaired PB1 extrusion, indicating that an appropriate level of Tfap2a was required for proper spindle assembly and chromosome alignment during mouse oocyte meiosis. 

Factors such as diabetes [12,13] and advancing maternal age [14,15] can affect meiosis and are thought to mainly result in aneuploidy. As a major global health threat, T2D is closely linked to the epidemic of obesity and polycystic ovary syndrome [18] and can lead to ovulatory dysfunction and infertility. Among numerous genes known to be associated with T2D risk, *TFAP2A* is important in driving expression of T2D genes [20,21]. After T2D induction, we observed significant spindle and chromosome defects in MII oocytes and high levels of Tfap2a in GV oocytes, suggesting a potential relationship between T2D and abnormal levels of Tfap2a.

To study the effects of Tfap2a-OE on oocyte maturation, we also performed an intracytoplasmic microinjection of Myc-Tfap2a mRNA into mouse GV oocytes. Understandably, there was only one band in the Tfap2a-OE group when the anti-Myc-tag antibody was used. Notably, when the anti-Tfap2a antibody was used, there was no band in the Myc group, but there were two bands in the Myc-Tfap2a group. To explain this phenomenon, we speculated that the endogenous Tfap2a levels of GV oocytes in the control group were too low to be detected by Western blotting, whereas the deep black bottom band in the Myc-Tfap2a group could have resulted from a large endogenous Tfap2a induction after Myc-Tfap2a overexpression. There are many transcription factors that can be induce by themselves, a phenomenon known as autoregulation. A transcription factor named PERICYCLE FACTOR TYPE-A (PFA) has the basic helix–loop–helix structure similar to that of Tfap2a. Overexpression of PFA caused ectopic expression of PFA in the root tip of *Arabidopsis* [62]. Overexpression of FOXL2 in KGN cells induced a significant enhancement of *FOXL2* transactivation [63]. By binding to the cis-elements within its promoter, the erythroid transcription factor *GATA-1* can positively regulate its expression in MEL cells [64]. Interestingly, a Tfap2a–binding site in the 2000 bp upstream promoter sequence of the *Tfap2a* gene has been predicted (data not shown), indicating an autoregulation of *Tfap2a*. However, the underlying mechanism requires further research.

Proper chromatin condensation was found to be vital for the subsequent fidelity of chromosome segregation into daughter cells [36,65]. Tfap2a-OE accelerated oocyte GVBD, resulting in an arrest of pro-MI/MI stage, indicating failed segregation of homologous chromosomes. Proper alignment of chromosomes on the metaphase plate guarantees equal separation of DNA [66,67], whereas disrupted spindle assembly and chromosomal alignment during meiosis I cause aneuploidy [68,69]. The end-on attachment of microtubules to kinetochores yields a bioriented interaction of chromosomes with the spindle, guaranteeing faithful chromosome segregation [70]. Centromere defects result in improper kinetochore attachments and improper chromosome segregation [71,72]. In the present study, a large amount of Tfap2a was located in the nucleus, with a small amount in the cytoplasm of GV oocytes in the Tfap2a-OE group, whereas in the Myc group, Tfap2a was mainly dispersed in the cytoplasm of GV oocytes. The expression priority of Tfap2a in the nucleus rather than in the cytoplasm is related to increased risk of dying [49,52]. We found that Tfap2a-OE predisposed oocytes to suffer from severely misaligned chromosomes and abnormal spindles at the MII stage (Figure 5A,B). As the chromosome alignment indicator, the MI-plate width was measured and found to be dramatically stretched after Tfap2a overexpression (Figure 5C). The above observations lead us to speculate that the aggregation in nucleus of Tfap2a after Tfap2a-OE yielded loosened chromatin, which might hinder the condensation of chromatin into chromosome, further causing centromere dysfunction; then, the kinetochore-microtubule attachment might be disrupted, and in turn, spindle assembly and chromosome alignment be affected, with mouse oocyte meiotic maturation eventually hindered.

To elucidate the effect of enhanced expression of Tfap2a in the nucleus, we detected a boosted p300 protein in the GV of oocytes, which can interact with Tfap2a to mediate transcriptional activation [27]. Given to the acetyltransferase property of p300, it was not difficult to draw the following inference: the elevated levels of Tfap2a in the nucleus of GV oocytes after Tfap2a-OE might have upregulated the p300 in the GV and ultimately contributed to the upregulated levels of histone acetylation.

Histone acetylation is reversible and associated with increased transcriptional activity in eukaryotic cells [39,40]. We suspected that Tfap2a-OE might cause chromatin loosening by promoting the acetylation of histone. Our results showed that levels of acetylation at pan and certain lysine sites of histone were significantly increased after Tfap2a overexpression. Additionally, elevated levels of H3K27 acetylation were observed via Western blotting (2.40 times higher than the control group; data not shown). These results collectively suggest that Tfap2a-OE can upregulate histone acetylation by elevating p300 in mouse GV oocytes.

As a newly discovered histone post-translational modification that is also catalyzed by p300 [30] and has been intensively studied in recent years, histone lactylation stimulates gene transcription from chromatin by loosening nucleosome–DNA interactions [73]. In recent years, only one article has focused on the dynamic changes in lactylation during oocyte meiosis, reporting that GV oocytes underwent lactylation in the pan histone, H3K23, and H3K18, whereas only pan histone lactylation and H3K23la occurred in the condensed chromosomes [34]. Our data indicated, for the first time, that as the chromatin configuration in GV oocytes transformed from NSN to SN (i.e., when growing oocytes transformed into fully grown oocytes), levels of pan histone lactylation, H3K18la, and H4K12la increased gradually (data in the control group), and levels of these lactylated sites elevated dramatically when Tfap2a was overexpressed. These results suggest that boosted p300 may be implicated in increased levels of histone lactylation after Tfap2a-OE in mouse GV oocytes.

## 4. Materials and Methods

### 4.1. Mice

Six-to eight-week-old ICR mice were purchased from the laboratory animal research center of the Xinglong Experimental Animal Breeding Factory (Beijing, China). Mice were fed ad libitum with a standard diet and were kept at a constant temperature in a light-controlled room (20–22 °C, 12:12 h light–dark), in accordance with the Animal Care Commission of the College of Animal Sciences and Technology, China Agricultural University.

### 4.2. Antibodies and Reagents

Rabbit anti-Tfap2a antibody (Cat# ab108311), rabbit anti-H4K16ac antibody (Cat# ab109463), and rabbit anti-p300 antibody (Cat# ab275378) were purchased from Abcam (Cambridge, UK); mouse anti-Myc-tag antibody (Cat# 2276) was purchased from Cell Signaling Technology; mouse anti-α-tubulin-CoraLite^®^594 antibody (Cat# CL594-66031) and rabbit anti-Histone H3 antibody (Cat# 17168-1-AP) were purchased from Proteintech (Wuhan, China); rabbit anti-H4K12ac antibody (Cat# A-4029-050) was obtained from Epigentek (Farmingdale, NY, USA); rabbit anti-Pan Lac antibody (Cat# PTM-1401RM), mouse anti Pan Ace antibody (Cat# PTM-102), rabbit anti-H3K18la antibody (Cat# PTM-1406RM), and rabbit anti-H4K12la antibody (Cat# PTM-1411RM) were purchased from PTM Bio Inc. (Chicago, IL, USA); Cy™3 AffiniPure goat anti-rabbit IgG (H+L) (Cat# 111-165-144), Cy™2 AffiniPure goat anti-mouse IgG (H+L) (Cat# 115-225-146), and Cy™2 AffiniPure donkey anti-rabbit IgG (H+L) (Cat# 711-225-152) were produced by Jackson ImmunoResearch Laboratory (West Grove, PA, USA). All other chemicals were purchased from Sigma Chemical Company (St. Louis, MO, USA), unless otherwise stated.

### 4.3. Induction of Type 2 Diabetes Mice (T2DM)

T2DM was generated as described previously [74]. Briefly, six-week-old ICR female mice (26 ± 2 g) were acclimated to the environment for one week. After acclimation, about half of the mice were fed with normal chow (3.42 kcal/g, 12% of energy from fat, Beijing Huafukang, 1022) and received an intraperitoneal injection of citrate buffer (pH 4.4–4.5, referred to as the control group). The other half were fed a high-fat diet (5.24 kcal/g, 60% of energy from fat, Beijing Huafukang, H10060) and received an intraperitoneal injection of streptozotocin (dissolved in a citrate buffer, 50 mg/kg, once per day for 7 consecutive days) to induce diabetes in week 9. After 7 consecutive days of injections, mice were fed for another 10 days. Subsequently, serum glucose levels were measured, and mice with blood glucose levels greater than 11.0 mM were used as diabetes mice for the following analyses.

### 4.4. Collection and Culture of Mouse Oocytes

Ovaries were removed 46–48 h after intraperitoneal injection of 7.5 IU pregnant mare serum gonadotropin (Shu Sheng Hormone, Cixi, China) and punctured in M2 medium to release the oocytes. COCs with more than three layers of unexpanded cumulus cells and containing oocytes with homogenous cytoplasm were selected and washed thoroughly with M2 medium under a dissecting microscope and cultured in α-MEM (Gibco-Invitrogen, Karlsruhe, Germany) containing 10% (*v*/*v*) fetal bovine serum (FBS, Gibco-Invitrogen, Karlsruhe, Germany) (with or without IBMX) under liquid paraffin oil at 37 °C in an atmosphere containing 5% CO_2_ in air. For denuded oocyte (DO) collection, COCs were stripped of cumulus cells by pipetting in M2 medium. Oocytes at the GV, GVBD, MI, and MII stages were collected for qRT-PCR and immunofluorescence staining after being cultured for 0, 3, 8, and 14 h, respectively.

### 4.5. Real-Time Quantitative-PCR (qRT-PCR)

RNA extracted from collected oocytes was preamplified using a single-cell sequence-specific amplification kit (P621, Vazyme Biotech Co., Ltd., Nangjing, China) according to the method described by Zheng et al. [75]. Briefly, 10 oocytes were added to 5 μL of a reaction mixture containing 2.5 μL 2 × reaction mix, 0.5 μL primer assay pool, 1.9 μL nuclease-free water, and 0.1 μL RT/Taq enzyme. The mixture was immediately incubated for 2 min at −80 °C, followed by centrifugation at 3000 rpm for 2 min at room temperature. Subsequently, the sample was incubated at 50 °C for 60 min and then at 95 °C for 3 min, followed by 17 cycles of 95 °C for 15 s and 60 °C for 15 min to complete the first round of amplification. qRT-PCR was performed using Taq Pro Universal SYBR qPCR Master Mix (Q712, Vazyme Biotech Co., Ltd., Nangjing, China) in a Bio-Rad CFX96 Touch™ system (Bio-Rad Laboratories, Inc. Hercules, CA, USA). The primers for amplification of the fragments of *Tfap2a* were as follows: forward: TAATGCCGACTTCCAGCCTC; reverse: AGGATTCAGGCTGTAGGGGT. *H2az1* was selected as the housekeeping gene using the following primers: forward: CGCAGAGGTACTTGAGTTGG; reverse: TCTTCCCGATCAGCGATTTG [76]. PCR conditions were as follows: after a 30 s incubation at 95 °C, amplification was performed for 40 cycles of 95 °C for 10 s and 60 °C for 30 s. Relative gene expression was calculated using the 2^−ΔCt^ method. The experiment was repeated at least three times using different sets of oocytes.

### 4.6. Cloning and Microinjection

Total cellular RNA from oocytes (100) was extracted using an RNAprep pure tissue kit (TIAN-GEN Biotech, Beijing, China); according to the manufacturer’s instructions, the first-strand cDNA was generated with a HiScript III 1st Strand cDNA Synthesis Kit (R312, Vazyme Biotech Co., Ltd., Nangjing, China) using Oligo(dT)_20_VN. We cloned the full length of *Tfap2a* cDNA using the following primers: forward: CCCAAGCTTATGAAAATGCTTTGGAAACTGAC; reverse: CGCGGATCCTCACTTTCTGTGTTTCTCTTCTTTG. The PCR products were purified with a gel extraction kit (DP214, TIAN-GEN Biotech, Beijing, China); then, the products were digested using HindIII and BamHI (New England Biolabs, Inc.) and cloned into the pCDNA3.1-3 × Myc-N vector (P1042, Miaoling Biology, Wuhan, China), in which the *Tfap2a* sequence was linked to 3 × Myc-tag at its C-terminus. After being transformed into *E. coli* DH5α, pCDNA3.1-3 × Myc-N and pCDNA3.1-3 × Myc-Tfap2a plasmids were then extracted (DP105, TIAN-GEN Biotech, Beijing, China) and used to validate the constructs via DNA sequencing. To generate mRNA for microinjection, the above two plasmids were linearized using restriction digest (XhoI) and purified via gel extraction. A T7 Message Machine Ultra Kit (AM1345, Ambion, Austin, TX, USA) was applied to produce capped mRNA, and the mRNA was then purified using a HiPure RNA Pure Micro Kit (R2144, Magentec, Guangzhou, China). The synthesized mRNA was dissolved in nuclease-free water and stored at −80 °C. For convenience, mRNA produced from pCDNA3.1-3 × Myc-N was renamed Myc, and that from pCDNA3.1-3 × Myc-Tfap2a was renamed Myc-Tfap2a.

### 4.7. Microinjection of mRNAs

Oocytes were microinjected as previously described [77] using an Narishige microinjector (Narishige Inc., Sea Cliff, NY, USA) with an Olympus IX70 microscope (Olympus, Center Valley, PA, USA) equipped with an Eppendorf TM FemtoJet TM 4i Microinjector (Eppendorf, Hauppauge, NY, USA) and completed within 30 min. For the overexpression experiments, 10 pL Myc-Tfap2a mRNA solution (1 μg/μL) was injected into the cytoplasm of denuded GV oocytes. The same amount of Myc mRNA (1 μg/μL) was injected as a control group. Following microinjection, the oocytes were arrested at the GV stage in α-MEM containing 10% (*v*/*v*) fetal bovine serum (Gibco-Invitrogen, Karlsruhe, Germany) and 200 nM IBMX for 12 h. Then, the oocytes were washed and cultured in an IBMX-free α-MEM medium (containing 10% (*v*/*v*) fetal bovine serum) under a humidified atmosphere of 5% CO_2_ at 37 °C for 14 h.

### 4.8. Western Blotting

Oocyte samples were prepared from groups of 300 DOs and lysed in a Laemmli buffer (Bio-Rad, Hercules, CA, USA) containing a protease inhibitor, followed by heating at 100 °C for 5 min. Then, the samples were separated using SDS-PAGE and transferred to a polyvinyl fluoride membrane (BioTraceTM NT; Pall Corp., FL, USA). The membranes were blocked for 1 h with 5% skim milk in TBS-T (with 0.1% Tween 20) at 37 °C and then probed with primary antibodies overnight at 4 °C (anti-Tfap2a antibody, 1:1000; anti-Myc antibody, 1:1000; anti-Histone H3 antibody, 1:5000). After washing three times in TBS-T (10 min each), the membranes were incubated at 37 °C for 1 h with HRP-conjugated anti-rabbit/mouse antibodies obtained from Zhongshan Biotechnology (Beijing, China). The protein bands were visualized using an Immobilon Western Chemiluminescent HRP substrate (SQ201; Shanghai EpiZyme Biotechnology, Inc., Shanghai, China), the images were scanned with a Tanon ultraviolet imaging system (Tanon-5200Multi, Shanghai, China), and data were analyzed using Image J 1.44 p software (National Institutes of Health, Bethesda, MD, USA).

### 4.9. Immunofluorescence Staining

Immunofluorescence staining was performed as described previously [78]. Briefly, oocytes were fixed in 4% paraformaldehyde for 1 h at 4 °C and permeabilized with 0.5% Triton X-100 for 45 min at 37 °C. After three washes in 0.1% polyvinyl alcohol (PVA)/PBS, the oocytes were blocked in 3% BSA-supplemented PBS for 1 h and incubated overnight at 4 °C with primary antibodies as follows: anti-Tfap2a antibody (1:200), anti-Myc antibody (1:500), anti-α-tubulin-CoraLite®594 antibody (1:200), anti-Pan Kac antibody (1:1000), anti-Pan Kla antibody (1:1000), anti-H4K12ac antibody (1:1000), anti-H4K16ac antibody (1:200), anti-H3K18la antibody (1:1000), anti-H4K12la antibody (1:1000), and anti-p300 antibody (1:200). After three washes in 0.1% PVA/PBS, the oocytes were incubated with Cy™3 AffiniPure goat anti-rabbit IgG (H + L) (1:1000), Cy™2 AffiniPure goat anti-mouse IgG (H+L) (1:200) or Cy™2 AffiniPure donkey anti-rabbit IgG (H+L) (1:200) for 1 h at room temperature. Then, the oocytes were further washed three times and stained with 10 µg/mL Hoechst 33,342 for 5 min in the dark. Finally, the oocytes were mounted on glass slides, and fluorescent imaging was performed using a Nikon A1HD25 confocal microscope (Minato, Tokyo, Japan). Fluorescence intensities and relative intensities were measured on the raw images using Image-Pro Plus software (Media Cybernetics Inc., Silver Spring, MD, USA) under fixed thresholds across all the slides. Both the fluorescence density and the area of the objects exhibiting the fluorescence were measured, and the mean relative intensity of fluorescence was calculated for each oocyte. For each histone modification site and p300, the average relative fluorescence intensity of the NSN oocytes from the Myc group was set to 1, and those from other treatment groups were expressed relative to this value. The average relative fluorescence intensity of Tfap2a in oocytes from the control group or the GV stage was set to 1, and those from other groups were expressed relative to this value.

### 4.10. Statistical Analysis

All the experiments were repeated at least three times unless otherwise specified. An unpaired Student’s t test was used for two-group comparisons. For comparison between multiple groups, data were subjected to a one-way analysis of variance (ANOVA) and Duncan’s test using the Statistics Package for the Social Sciences (SPSS 18.0, SPSS Inc. Chicago, IL, USA). All data are expressed as the mean ± SEM (standard error of the mean). *p* values less than 0.05 were used to determine significant differences. * *p* < 0.05, ** *p* < 0.01, *** *p* < 0.001, and **** *p* < 0.0001.

## 5. Conclusions

In conclusion, our results suggest that Tfap2a overexpression might have upregulated p300, increased levels of histone acetylation and lactylation, impeded spindle assembly and chromosome alignment, and ultimately hindered mouse oocyte meiosis. There seems to be a correlation between high levels of Tfap2a in GV oocytes and harm to oocyte quality caused by T2D, although the underlying mechanism requires further research. If the mechanism is clarified, the protein is expected to be applied as a marker to predict the quality of oocytes in people with T2D and become a target protein for treatment of the disease.

## Figures and Tables

**Figure 1 ijms-23-14376-f001:**
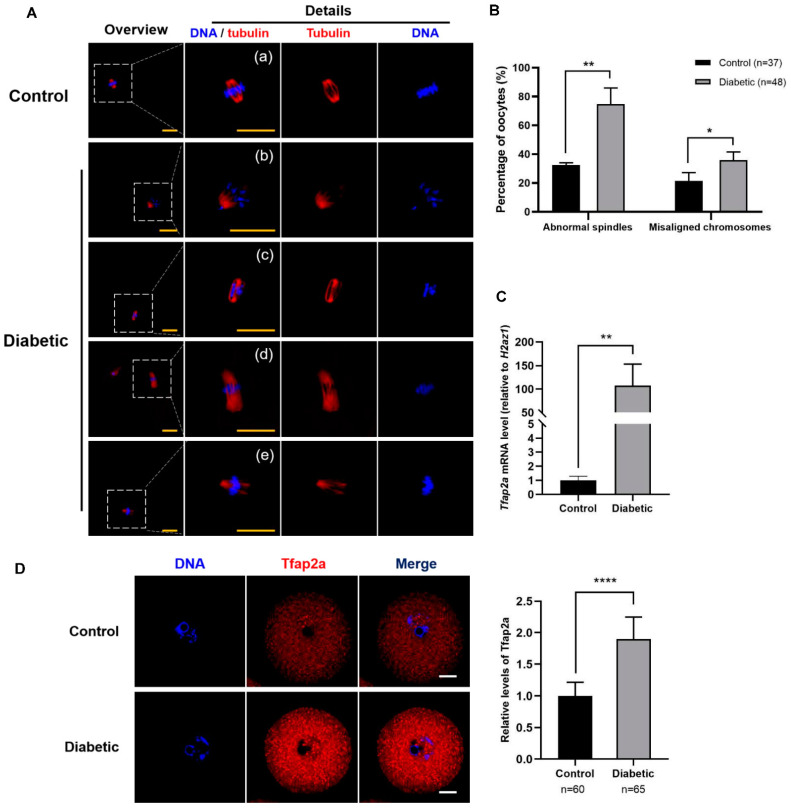
Maternal diabetes-induced spindle defects and chromosome misalignment in MII oocytes and a high level of Tfap2a in GV oocytes. (**A**) Spindle and chromosome morphologies in MII oocytes after in vitro maturation for 14 h in the control and diabetic mice. Most oocytes in the control group showed normal spindle and chromosome alignment (a), whereas in the diabetes group, oocytes revealed asymmetrical (b), elongated (d), and multipolar (c and e) spindles, and some chromosomes moved to the poles of the spindle (b) or clustered on the metaphase plate (c). Magnifications of the boxed regions are shown on the right side of each main panel. Tubulin is shown in red, and DNA is indicated in blue. Scale bar = 20 µm. (**B**) Quantification analysis of MII oocytes with spindle defects and chromosome misalignment from control or diabetic mice. The data are expressed as the mean ± SEM of at least three independent experiments. * *p* < 0.05, ** *p* < 0.01. (**C**) Relative mRNA expression of *Tfap2a* in GV oocytes from the control and diabetic mice. Levels of expression were normalized to levels of *H2az1*, and each bar represents the mean ± SEM (*n* = 4). ** *p* < 0.01. (**D**) Images of the subcellular localization and relative intensity of Tfap2a in GV oocytes from the control and diabetic mice. Tfap2a is shown in red, and DNA is indicated in blue. Scale bar = 20 µm. Data are represented as the mean ± SEM of three independent experiments. **** *p* < 0.0001.

**Figure 2 ijms-23-14376-f002:**
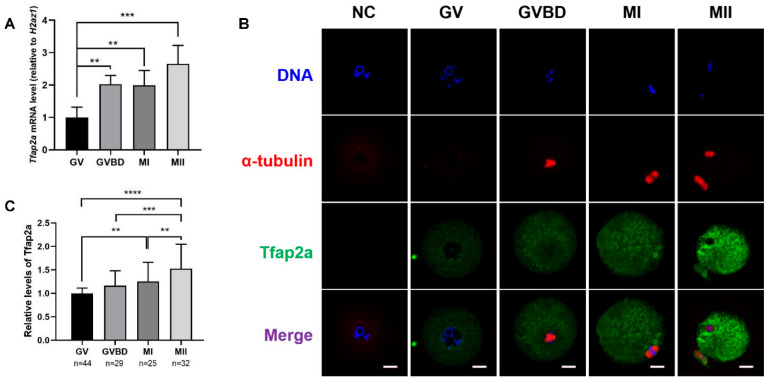
The levels and subcellular localizations of Tfap2a at different stages during mouse oocyte maturation. (**A**) The levels of *Tfap2a* mRNA at different stages based on qRT-PCR analysis. A total of 10 oocytes were collected after culturing for 0, 3, 8, or 14 h, corresponding to GV, GVBD, MI, and MII stages in each repeat, respectively. The levels of mRNA expression were normalized to *H2az1*, and each bar represents the mean ± SEM (*n* = 4). ** *p* < 0.01, *** *p* < 0.001. (**B**) The subcellular localizations of Tfap2a at different stages based on immunofluorescence. α-tubulin (red), DNA (blue), Tfap2a (green). NC: negative control; scale bar = 20 µm. (**C**) Relative fluorescence intensity of Tfap2a at different stages. Data are represented as the mean ± SEM of three independent experiments. ** *p* < 0.01, *** *p* < 0.001, **** *p* < 0.0001.

**Figure 3 ijms-23-14376-f003:**
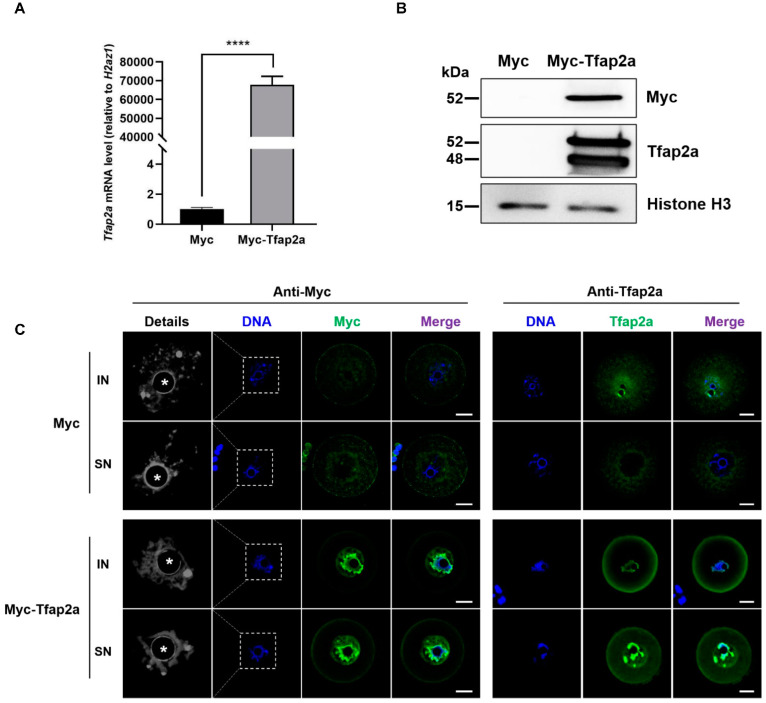
Overexpression of Tfap2a loosened chromatin in mouse oocytes. (**A**) The levels of *Tfap2a* mRNA between Myc and Myc-Tfap2a mRNA-microinjected oocytes. The GV-stage oocytes were microinjected with Myc or Myc-Tfap2a mRNA, respectively, and incubated with 200 nM IBMX for 12 h for qRT-PCR. Each sample contained 10 oocytes. The levels of expression were normalized to *H2az1*, and each bar represents the mean ± SEM (*n* = 3). **** *p* < 0.0001. (**B**) The levels of Tfap2a in oocytes between Myc and Myc-Tfap2a mRNA microinjection via Western blotting. Myc-Tfap2a: 300 oocytes injected with 1 μg/μL Myc-Tfap2a mRNA solution; Myc: 300 oocytes injected with the same amount of Myc mRNA. The top band shows Myc level based on the antibody against Myc-tag. Only one band was detected in the Myc-Tfap2a lane, and the molecular mass of Myc-Tfap2a was approximately 52 kDa. The middle of the figure presents Western blotting, showing Tfap2a expression detected via the antibody against Tfap2a. In the Myc-Tfap2a group, two bands were detected. The molecular mass of the three Myc tags was about 3.6 kDa. The 52 kDa molecular mass was that of the Myc-Tfap2a fusion protein, and the 48 kDa molecular mass was that of the endogenous Tfap2a protein. Histone H3 served as a loading control with a size of approximately 15 kDa. (**C**) The subcellular localizations of Myc and Myc-Tfap2a fusion protein in the IN- and SN-type mouse oocytes via immunofluorescence staining. Magnifications of the boxed regions are shown on the left of each main panel (pseudo-colored in greyscale). Myc and Tfap2a are shown in green, and DNA is indicated in blue. Nucleoli are indicated with asterisks (*). Scale bar = 20 µm.

**Figure 4 ijms-23-14376-f004:**
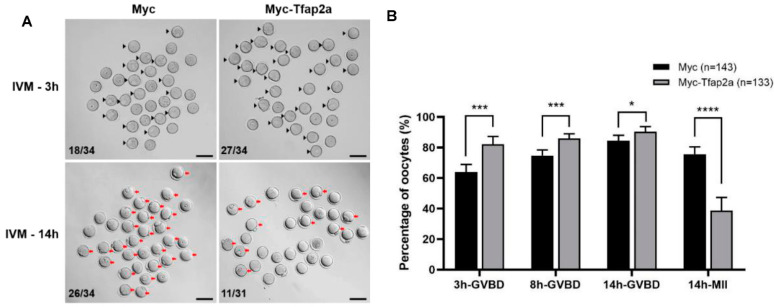
Overexpression of Tfap2a promoted GVBD but inhibited extrusion of the first polar body in mouse oocytes. (**A**) Images of Myc and Myc-Tfap2a mRNA-injected oocytes at different times after maturation in vitro. The arrowheads indicate the GVBD oocytes 3 h after IVM, whereas the red arrows indicate the oocytes with PB1, and the numbers in the images indicate the ratios of GVBD or PB1 extrusion to the total number of oocytes. Scale bar = 100 μm. (**B**) The effects of Tfap2a overexpression on the rates of GVBD and PB1 oocytes at different times after IVM. Data are represented as the mean ± SEM of at least three independent experiments. * *p* < 0.05, *** *p* < 0.001, **** *p* < 0.0001.

**Figure 5 ijms-23-14376-f005:**
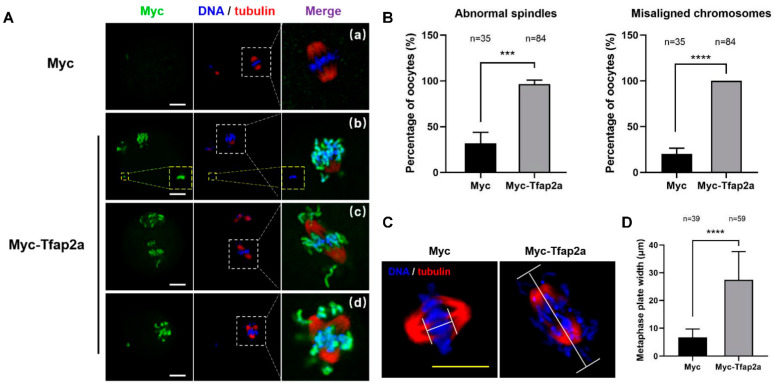
Overexpression of Tfap2a caused abnormal spindles and misaligned chromosomes in mouse oocytes. (**A**) Abnormal spindle and chromosome alignment in MII oocytes after the microinjection of Myc-Tfap2a mRNA. In the Myc group, most oocytes showed normal spindle and chromosome alignment (a), whereas in the Myc-Tfap2a group, Myc-Tfap2a protein was colocalization with nuclear DNA, and the chromosomes were severely misaligned in most oocytes. Some chromosomes moved to the poles of the spindle or were largely scattered in the cytoplasm (b and c), and the spindles showed unattached microtubules (b), which were severely disrupted or multipolar (d). Magnifications of the boxed regions are shown on the right of each main panel. Myc is shown in green, tubulin in red, and DNA in blue. Scale bar = 20 µm. (**B**) The rates of oocytes with abnormal spindles or misaligned chromosomes after overexpression of Myc or Myc-Tfap2a. The data are expressed as the mean ± SEM of at least three independent experiments. *** *p* < 0.001, **** *p* < 0.0001. (**C**) The MI oocytes stained by anti-tubulin (red) and Hoechst 33342 (blue). Scale bars: 20 μm. (**D**) The width of the metaphase plate measured by the axis-to-axis distance between the two lines at the edges of the DNA in (**C**). The data are expressed as the mean ± SEM of at least three independent experiments. **** *p* < 0.0001.

**Figure 6 ijms-23-14376-f006:**
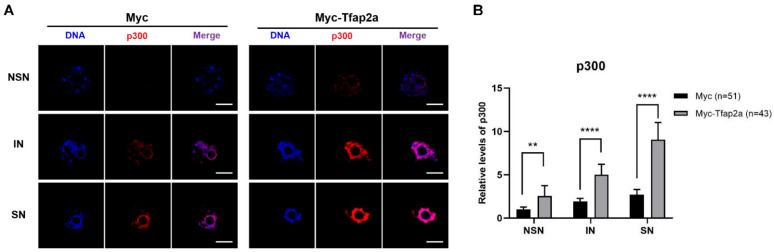
Subcellular localizations and levels of p300 after the overexpression of Tfap2a. (**A**) GV-stage oocytes were microinjected with Myc or Myc-Tfap2a mRNA, respectively, and incubated with 200 nM IBMX for 12 h. p300 is shown in red, and DNA is indicated in blue. Scale bar = 20 µm. (**B**) Quantification data of images in (**A**). The average relative fluorescence intensity of the NSN oocytes from the Myc group was set to 1. Data are expressed as the mean ± SEM of at least three independent experiments. ** *p* < 0.01, **** *p* < 0.0001.

**Figure 7 ijms-23-14376-f007:**
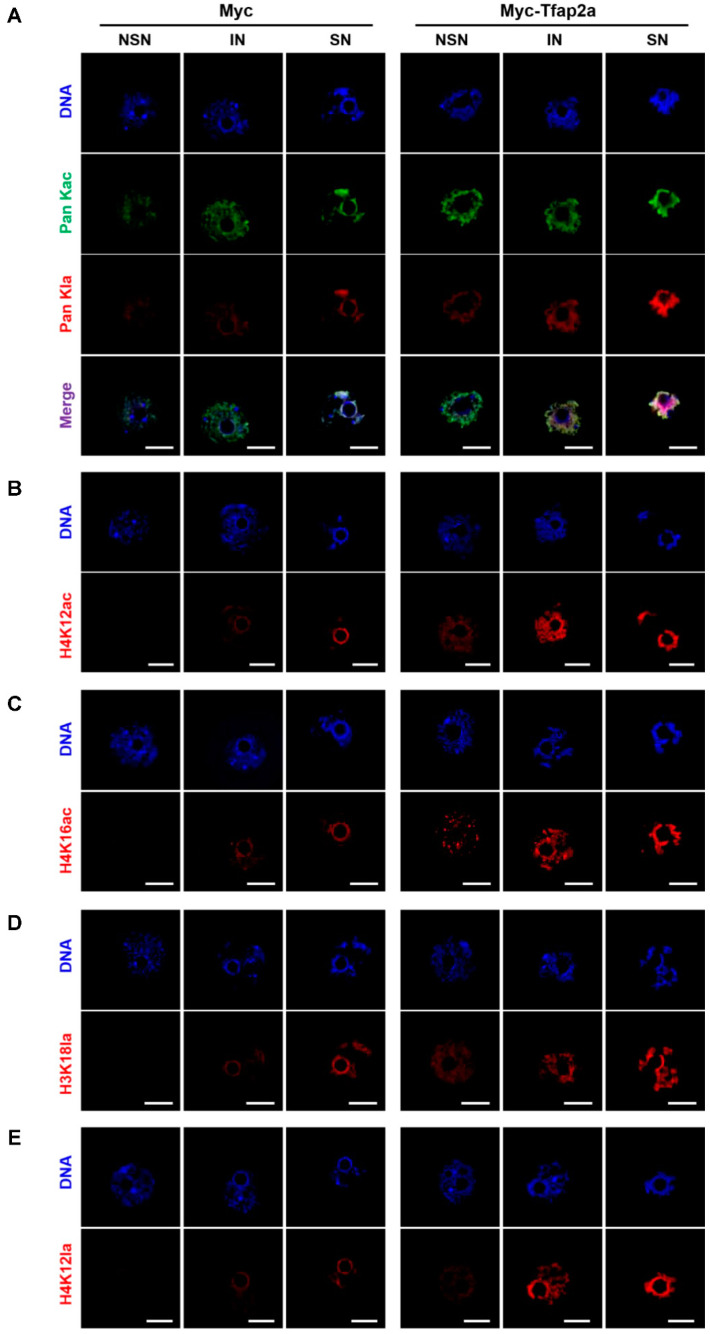
Representative pictures of histone acetylation and histone lactylation in mouse GV-stage oocytes after overexpression of Myc and Myc-Tfap2a. The pan histone acetylation and the pan histone lactylation (**A**), H4K12ac (**B**), H4K16ac (**C**), H3K18la (**D**), and H4K12la (**E**) in mouse oocytes at the GV stage. Pan histone acetylation is shown in green, whereas pan histone lactylation, H4K12ac, H4K16ac, H3K18la, and H4K12la are shown in red. The DNA was stained with Hoechst 33342 (blue). Scale bars = 20 μm.

**Figure 8 ijms-23-14376-f008:**
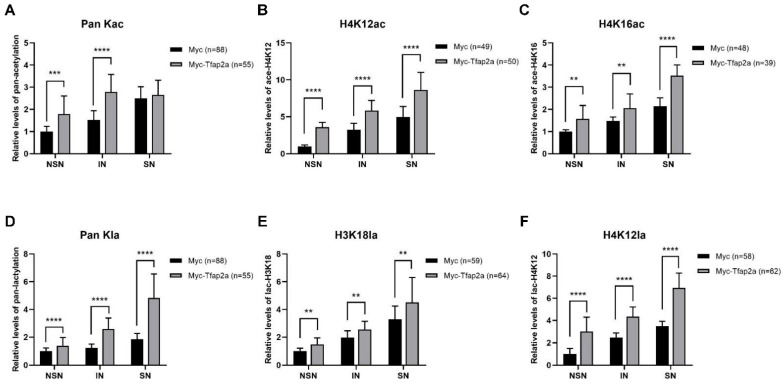
Quantification of fluorescence intensities of histone acetylation and lactylation at different phases in mouse GV oocytes after the overexpression of Myc and Myc-Tfap2a shown in Figure 7. The fluorescence intensities of pan histone acetylation (**A**), H4K12ac (**B**), H4K16ac (**C**), pan histone lactylation (**D**), H3K18la (**E**), and H4K12la (**F**) were quantified based on immunofluorescence staining. The average relative fluorescence intensity of the NSN oocytes from the Myc group was set to 1. The data are expressed as the mean ± SEM of at least three independent experiments. ** *p* < 0.01, *** *p* < 0.001, **** *p* < 0.0001.

## Data Availability

The data that support the findings of this study are available from the corresponding author upon reasonable request.

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
