# Peer review of "Overexpression of Tfap2a in Mouse Oocytes Impaired Spindle and Chromosome Organization"

_ijms, 2022, doi:10.3390/ijms232214376_

Round 1
Reviewer 1 Report
In this study the authors analyse the relevance of the transcription factor Tfap2a in the oogenesis development. They have overexpressed this transcription factor and studied the effect in the oogenesis progression and in the acetylation and lactylation of some histones. The experiments have been well done and the results are consistent and very interesting. I have minor issues that I believe could improve the paper.
Figures:
- The scale bars in almost all the figures are very difficult to be distinguished, maybe due to the red colour. In my opinion, it would be better if the scale bars were white and a little thicker.
- The colour blue in which we can see DNA is not very well appreciated, so some of the cytological descriptions are difficult to appreciate. For example, it is difficult for me to see the Chromatin Loosen Caused by Tfap2a overexpression. Maybe, if the DNA images were pseudo-coloured in a brighter blue or just in grey scale it would be easier to detect the cytological changed.
-In figure 4A, the yellow lines in row b enlarged Myc signal, but the small square is into the DNA/tubulin column. I think it would be better if it was placed in the Myc row.
- I figure 8A the images are so small that it is very difficult to appreciate the differences. The details should be enlarged a little more
Results
-In line 132 it is referred fig 2B while I think it should be referred the figure 2C
-How many oocytes were injected and analysed with Myc or Myc-Tfap2a for the experiment showed in figure 5? Probably it is my fault, but I can´t found this information.
Discussion
The authors claimed that “These results collectively suggested that Tfap2a-OE was able to disrupt spindle assembly and chromosome alignment, thus hindering mouse oocyte meiotic maturation” (lines 317, 318). But this overexpression is really affecting the spindle organization, or it is just a consequence of the chromatin and meiotic process alteration. Is it possible that some relevant chromosome structures, such as centromeres, are being affected? Perhaps there could be a little more discussion on this point.
Reviewer 2 Report
The manuscript ‘Tfap2a Overexpression Disrupts Mouse Oocytes Meiotic Maturation by Impairing Spindle/chromosome Organization Through p300 Mediated Acetylation and Lactylation of Histones’ describes the effect of Transcription factor AP-2-alpha overexpression on the chromatin organization and epigenetic histone marks. Authors revealed that overexpression of Tfap2a increased levels of histone acetylation and lactylation, inhibited spindle assembly and chromosome alignment during mouse oogenesis.
Authors represent high quality original data that is clearly summarized in 8 main figures. The manuscript is logically written and the data for the most part are convincing. However the following questions should be addressed to confirm the conclusions drawn:
Q1: The title of the manuscript. It is not directly shown in the manuscript that histone acetylation and lactylation changes through p300 mediated process. Only correlation is demonstrated and this should be reflected in the title of the manuscript.
Q2: It is unclear from the Introduction why Tfap2a was chosen for the experiments, why histones (and their particular modifications, including lactylation, were analyzed), why diabetes model was used. A clear description of the aims of the work is required. In accordance with this clarified goal, the structure and the order of the presentation of the results could be changed, if necessary.
Q3: In the discussion section, a more detailed description about the relationship between diabetes, oocytes, chromosomes, histones, Tfap2a, and the results obtained is needed.
Q4: “Oocytes were directly added to a reaction mixture” – a separate control is required to demonstrate that (1) PCR inhibition did not occur and (2) RNA was not degraded by RNases which possess ability to renaturate after heating.
Q5: H2az1 used for normalization:
- It is necessary to check that the level of H2az1 transcript does not change under experimental conditions;
- Since each PCR reaction contains equal number of oocytes, no reference gene normalization is needed; dCt method can be used instead of ddCt;
- qPCR data usually does not follow normal distribution, so non-parametric statistical test is required (not Student t test).
Q6: Verification of Tfap2a protein expression after microinjection is a key methodological step in the work, therefore, authors should find out and provide a detailed description of why:
- Endogenous Tfap2a was not detected by western-blotting in control oocytes (fig 2b);
- In Myc-Tfap2a group two bands were detected with anti-Tfap2a antibodies in contrast to control (fig 2b).
Subcellular localization of Myc-Tfap2a fusion protein should be also analysed with anti- Tfap2a antibodies (fig 2c).
Q7: Levels of histone modifications and level of p300 were estimated according to immunofluorescence staining. This method should be accompanied by another more direct quantitative approach such as western-blotting. Moreover, quantification of fluorescence intensity after oocyte imaging should be described in detail in the Methods section.
Minor comments:
C1: Avoid repeats in text (methods in the Results section).
C2: 3.1 qRTPCR “The mixture was immediately incubated for 2 min at -80 °C, followed” – minus 80 °C?
C3: “4.6. mRNA Synthesis and Microinjection” should be changed to “Cloning and microinjection”
Round 2
Reviewer 2 Report
The authors took into account all my comments and suggestions, which helped to improve the manuscript. The manuscript can be accepted for publication after minor corrections:
C1. Please consider changing the title of the manuscript to “Overexpression of Tfap2a in Mouse Oocytes Impaired Spindle and Chromosome Organization”.
C2. "Immunofluorescence staining hybridized with anti-Myc antibody" - please rephrase to avoid term "hybridize" to decsribe antibody – protein interactions. Replace it with “interacts” “reacts” or “detects”.
